# Constitutive Equation and Hot Processing Map of Mg-16Al Magnesium Alloy Bars

**DOI:** 10.3390/ma13143107

**Published:** 2020-07-12

**Authors:** Zongwen Ma, Fengya Hu, Zhongjun Wang, Kuijun Fu, Zhenxiong Wei, Jiaji Wang, Weijuan Li

**Affiliations:** 1School of Materials and Metallurgy, University of Science and Technology Liaoning, Anshan 114051, China; ustlmzw@gmail.com (Z.M.); zhenxiongwei111@gmail.com (Z.W.); ustllwj@gmail.com (W.L.); 2State Key Laboratory of Metal Material for Marine Equipment and Application, Ansteel Group Corporation, Anshan 114009, China; aghfy312@gmail.com (F.H.); agfkj63@gmail.com (K.F.); agwjj2013@gmail.com (J.W.)

**Keywords:** magnesium alloy, Mg-16Al, hot deformation, constitutive equation, processing maps

## Abstract

A Gleeble-2000D thermal simulation machine was used to investigate the high-temperature hot compression deformation of an extruded Mg-16Al magnesium alloy under various strain rates (0.0001–0.1 s^−1^) and temperatures (523–673 K). Combined with the strain compensation Arrhenius equation and the Zener–Hollomon (*Z*) parameter, the constitutive equation of the alloy was constructed. The average deformation activation energy, *Q*, was 144 KJ/mol, and the strain hardening index (n ≈ 3) under different strain variables indicated that the thermal deformation mechanism was controlled by dislocation slip. The Mg-16Al alloy predicted by the Sellars model was characterized by a small dynamic recrystallization (DRX) critical strain, indicating that Mg_17_Al_12_ particles precipitated during the compression deformation promoted the nucleation of DRX. Hot processing maps of the alloy were established based on the dynamic material model. These maps indicated that the high Al content, precipitation of numerous Mg_17_Al_12_ phases, and generation of microcracks at low temperature and low strain rate led to an unstable flow of the alloy. The range of suitable hot working parameters of the experimental alloy was relatively small, i.e., the temperature range was 633–673 K, and the strain rate range was 0.001–0.1 s^−1^.

## 1. Introduction

Magnesium (Mg) alloys are characterized by excellent specific strength, specific stiffness, electromagnetic shielding, damping, and other characteristics, and they have good application prospects in many fields (e.g., automobiles, electronic communications, and aerospace) [1,2,3]. The unique close-packed hexagonal crystal structure of Mg alloys and the limited sliding system at ambient temperature limit the workability and, hence, industrial application of these alloys [4]. Therefore, optimizing the thermal processing parameters and mastering the thermal deformation characteristics of Mg alloys are essential for controlling the structural evolution and mechanical properties of these materials. A thermal processing map based on the dynamic material model (DMM) combined with the microstructure of the material is used to optimize the thermal processing parameters of the material. For Mg-Al alloys, adding the Al element to Mg is effective in improving the yield strength and casting performance of a Mg alloy. The work of Prasad et al. [5] suggested that homogenization treatment is beneficial for expanding the processable area of cast AZ31 Mg alloy and leads to a significant reduction in the number of intergranular cracks and area of rheological instability. Wei et al. [6] discussed the hot tensile deformation characteristics and obtained the hot working map of an aged and homogenized AZ61Ce Mg alloy sheet. The results indicated that the aging treatment and homogenization treatment were beneficial for expanding the workable area of the alloy. Lou et al. [7] found that the processing map of extruded AZ80 Mg alloy in the stretched state consisted of two dynamic recrystallization (DRX) regions with small equiaxed crystals. Xu et al. [8] reported that the DRX tissue region of AZ91D exhibited good thermal processing performance, and this region was taken as the processable region for AZ91D thermal processing.

The occurrence of DRX will generally promote grain refinement and reduce the flow stress, thereby helping to improve the formability of materials. Furthermore, the relatively low stacking fault energy (SF) and high grain boundary diffusion rate of Mg alloy are beneficial to the occurrence of DRX [9]. Poliak and Jonas [10] proposed that an inflection point will occur in the strain hardening rate (*θ*) and stress (*σ*) curves of materials undergoing DRX. Poliak et al. [10,11,12,13] used a second derivative method to predict the critical strain (*ε_c_*) of DRX. The Sellars model [14] is a prediction model based on the Poliak–Jonas criterion, where the influence of temperature (T) and strain rate (*έ*) is considered, and the Zener–Hollomon (*Z*) parameter is used to obtain the *ε_c_* of DRX. The Sellars models of Mg-11.95Gd-4.5Y-2Zn-0.37Zr, AZ80, and AZ61 alloys were established by Yu [15], Su [16], and Wei [17], respectively. 

The Arrhenius-type constitutive equation has been widely used to describe the changing characteristics of flow stress and deformation temperature and strain rate [18,19,20,21]. However, only a few reports have considered the high temperature plastic deformation behavior of deformed Mg alloys with high Al content. Therefore, in the present work, the hot compression deformation characteristics of Mg-16Al Mg alloy hot extruded bars were investigated, and the corresponding constitutive equation and hot processing map were established. The optimal hot working parameter range of the alloy was determined, which provided a basis for optimizing the development of Mg alloys with high Al content.

## 2. Materials and Methods 

A semi-continuously cast Mg alloy bar with a diameter of 135 mm was cold-cut and peeled to a diameter of 120 mm. This bar was then hot-extruded at 673 K into a 45 mm–diameter bar, which was taken as the experimental material. After hot extrusion, the material was left to cool in the air. The nominal chemical composition of the material was 16 wt.% Al, 0.2 wt.% Zn, 0.3 wt.% Mn, 0.4 wt.% Ce, 0.2 wt.% Sr, and balance Mg. Prior to hot compression deformation, the experimental alloy was solution-treated at a temperature of 673 K for 12 h. As shown in Figure 1, the hot compression sample was taken at a position 0.6 R away from the axial center of the test bar and processed into a Φ12.0 × 18.0 mm cylinder.

The specimens after solution treatment were subjected to a thermal compression test on a Gleeble-2000D thermal simulation test machine (Data Sciences International, DE, USA), where the compression axis was set parallel to the existing extrusion direction. Temperatures of 523, 573, 623, and 673 K, and strain rates of 0.0001, 0.001, 0.01, and 0.1 s^−1^, respectively (maximum engineering strain: 80%), were employed during the experiment. A small amount of lubricating oil was inserted between the indenter and the specimen, to reduce the friction between these components. Each specimen was heated to the set temperature, held for 3 min, and then subjected to the hot compression test. After testing, the specimen was immediately cooled in water, to retain the structural characteristics after thermal deformation.

The microstructure of the Mg-16Al Mg alloy was examined via metallographic observations. For this examination, the specimens after hot compression deformation were cut along the direction parallel to the compression axis, and the section microstructures were observed. The specimens were polished and etched for optical microscopy (OM) observation. An etchant (1 mL water, 1 mL glacial acetic acid, 10 mL absolute ethanol, and 0.5 g picric acid solution) was used in the preparation of the samples. Cotton was dipped into the solution, which was then lightly smeared in one direction along the polished surface of the specimen. After a certain corrosion time, the surface was wiped with cotton, washed with water, and then washed with alcohol, to remove any residual corrosive solution. After drying, the microstructures of the specimens were observed under an optical microscope, and the fine microstructures were observed by scanning electron microscope (SEM, Zeiss-ΣIGMA HD, Thuringia, Germany) equipped with energy dispersive spectrometer (EDS).

## 3. Results and Discussion

### 3.1. The Original Microstructure of Mg-16Al before Deformation

The specimens subjected to a hot extrusion and solution treatment (673 K, 12 h) were observed via SEM. As shown in Figure 2a, the original hot extrusion microstructure was mainly composed of matrix *α*-Mg grains, *β*-Mg_17_Al_12_ phases with discontinuous reticular distribution near the *α*-Mg grains boundaries (GBs), and *γ*-Mg_17_Al_12_ phases with a cellular structure inside the *α*-Mg grains. Moreover, unevenly distributed Ce-containing rare-earth phases were also present in the hot extrusion specimens. The *γ*-Mg_17_Al_12_ (i.e., a cellular phase, as previously stated) is shown in Figure 3. Figure 2b shows the microstructure after the solution treatment. The massive *α*-Mg grains in the specimens are considerably larger than those in the hot-extruded specimens. Coarse *β*-Mg_17_Al_12_ phases at the GBs were only partly dissolved in the matrix, owing to the large number of Al atoms in the experimental alloy. Due to the solution treatment, the previously discontinuous reticular *β*-Mg_17_Al_12_ phases were partially dissolved, whereas the *γ*-Mg_17_Al_12_ phases were completely dissolved in the *α*-Mg matrix. Similarly, Al-atom diffusion into the matrix accompanied by grain boundary migration led eventually to an increase in the size of the *α*-Mg grains (average size increased from 10 to 25 μm).

### 3.2. The Microstructure of Mg-16Al after Deformation

Figure 4a shows a SEM image of the experimental alloy after deformation at low temperature and strain rate (523 K/0.0001 s^−1^). The microstructure consisted of many uniformly distributed recrystallized grains (average grain size: 15 μm). Numerous Al atoms diffused into the *α*-Mg grains, and, hence, many *γ*-Mg_17_Al_12_ phases were continuously precipitated in the final grains. Moreover, discontinuously distributed *β*-Mg_17_Al_12_ phase particles were present at the GBs. Figure 4b shows an SEM image of the experimental alloy after deformation at high temperature and strain rate (673 K/0.1 s^−1^). The alloy underwent DRX, which yielded an average grain size of 10 μm. The discontinuously distributed *β*-Mg_17_Al_12_ phase particles at the GBs were stretched along the direction rotated 45° with respect to the compression axis. Similarly, a small number of Mg_17_Al_12_ phases were discontinuously precipitated from the *α*-Mg matrix at the GBs.

### 3.3. Flow Behavior

The stress–strain curves of the experimental alloy specimens after solution treatment at various deformation conditions are shown in Figure 5. Obviously, the flow stress was affected by temperature and strain rate, and the peak stress and steady-state stress both increased with decreasing temperature and increasing strain rate. The stress–strain curves all exhibited significant recrystallization characteristics and a consistent variation trend: The flow stress increased sharply with increasing true strain, and then the rate of increase decreased gradually. The stress decreased gradually after reaching a peak value. When the deformation reached a certain true strain, the stress value remained basically unchanged. Moreover, the work hardening rate and flow softening rate varied with the strain rate and deformation temperature. At relatively high strain rate levels ($\dot{\varepsilon }$ ≥ 0.01 s^−1^), considerable work hardening and subsequent continuous flow softening were observed, but a significantly different flow behavior was observed at lower strain rates. At relatively high temperatures (T ≥ 623 K), the initial work hardening component decreased, and the flow softening appeared to be steady. The initial work hardening occurred at relatively low strain levels when temperatures lower than 623 K were employed. This hardening was followed by mild flow softening, and a dynamic balance between work hardening and softening was eventually reached. In addition, a comparison of the stress–strain curves with those of other AZ series commercial Mg alloys [22] revealed that the peak strain (*ε_p_*) corresponding to the peak stress of the experimental alloy was small.

### 3.4. Constitutive Analysis 

Research on different materials reveals that thermal deformation is a process controlled by thermal activation. The deformation behavior was greatly affected by strain rate and deformation temperature. According to the characteristics of the flow stress–strain curves, the relationship between flow stress and deformation conditions (temperature and strain rate) can be expressed by the hyperbolic sine Arrhenius-type equation proposed by Sellars [23]: (1)$$\dot{\varepsilon }=A{\left[\sinh\left(\alpha \sigma \right)\right]}^{n}\exp\left(-\frac{Q}{RT}\right)$$
where $\dot{\varepsilon }$ is the strain rate, *A* (s^−1^) and *α* (MPa^−1^) are the material constants, *σ* is the flow stress (MPa), *n* is the stress index, *R* is the general gas constant (8.314 J·mol^−1^K^−1^), T is the absolute temperature (K), and *Q* is the activation energy of thermal deformation (KJ/mol). Equation (1) can be expressed as two power-law expressions, which are expressed as Equations (2) and (3), under low-stress and high-stress conditions, respectively [24]:(2)$$\dot{\varepsilon }=A_{1}\sigma^{n_{1}}\exp\left(-\frac{Q}{RT}\right)$$
(3)$$\dot{\varepsilon }=A_{2}\exp(\beta \sigma )\exp\left(-\frac{Q}{RT}\right)$$
where *A*_1_, *A*_2_, *β* = *αn*_1_ are material constants. The maximum stress (peak stress *σ_p_*) in the flow curve can be used as the representative stress of each flow curve [25]. To study the flow characteristics of Mg-16Al Mg alloy during hot working after solution treatment, the peak stress, *σ_p_*, was used to calculate the constitutive equation parameters in this work. After taking the natural logarithms for both sides of Equations (1)–(3), it is found that *n*, *n*_1_, and *β* are the slopes obtained from the linear fitting results of the curves of ln$\dot{\varepsilon }$ versus ln*σ*, *σ*, and ln[sinh(*ασ*)], respectively, as shown in Figure 6 and Figure 7a. The *n*_1_, *β*, *n*, and *α* values of the experimental alloys under various deformation conditions were 4.437, 0.084, 2.840, and 0.0189, respectively.

According to Equation (1), the deformation activation energy *Q* can be defined as
(4)$$Q=R{\left[\frac{\partial \ln\dot{\varepsilon }}{\partial \ln[\sinh(\alpha \sigma )]}\right]|}_{T}\cdot {\left[\frac{\partial \ln[\sinh(\alpha \sigma )}{\partial (1/T)}\right]}_{\dot{\varepsilon }}=RnS$$
where the stress index, *n*, is the average slope of the ln$\dot{\varepsilon }$-ln[sinh(*ασ*)] linear fitting curves at a certain temperature, and S is the average slope of the ln[sinh(*ασ*)]-1/T linear fitting curves at a certain strain rate. According to the linear fitting results of Figure 7a,b, the average activation energy *Q* of Mg-16Al Mg alloy under different deformation conditions can be calculated to be 143.99 KJ·mol^−1^.

Additionally, the constitutive equation parameters (*Q*, *α*, and *n*) obtained for different types of AZ-based deformed Mg alloys [22] under various deformation conditions were summarized, as shown in Table 1. The *Q* of the experimental alloy was greater than the values of AZ41, AZ61, and AZ80. Moreover, this value was far greater than the activation energy of grain boundary self-diffusion in Mg (92 kJ/mol) [26], which was higher than that of lattice self-diffusion energy in Mg (135 kJ/mol) [27]. However, the *Q* value was close to the diffusion activation energy of Al in Mg (143 KJ/mol), indicating that the rate control mechanism of the alloy during the initial deformation was solute diffusion [28]. The *Q* values of AZ41, AZ61, and AZ80 decreased with increasing Al content. The reason was that the *β*-Mg_17_Al_12_ phase particles become softened at temperatures exceeding 423 K [29], and the second phase particles at the GBs weaken the obstacles to dislocation motion, thereby reducing the deformation activation energy of the alloy. However, for the experimental alloy, the relatively high deformation activation energy may be explained in terms of two effects. On the one hand, the solution treatment was performed prior to the plastic deformation, but many of the Al atoms were only partly dissolved in the alloy matrix. To a certain extent, the barrier effect of *β*-Mg_17_Al_12_ phases at the GBs and the Mg_17_Al_12_ phases precipitated from the matrix during the deformation was weakened. The number of particles in the second phase was more than that of other alloys, and the inhibition effect was negligible. On the other hand, numerous Al atoms dissolved in the matrix increased the hindrance to dislocation motion, thereby increasing the energy required for dislocation cross slip and climb.

A different stress index, *n*, reflects different creep mechanisms. When *n* is 2, the grains can rotate with each other, the grain boundaries can be coordinated, and slip can occur; when *n* is 3, the grains of the matrix larger, with fewer and coarse grain boundaries, solute dragging the grain boundaries will occur, at which time the dislocation slip creep becomes the main deformation mechanism; when *n* is 5, the creep mechanism is the dislocation creep controlled by climbing [30]. The *n* value of Mg-16Al Mg alloy was approximately equal to 3 under different strain conditions, which was lower than that of other AZ-based Mg alloys, indicating that the deformation mechanism of the alloy during deformation was controlled by dislocation slip.

The influence of temperature and strain rate on the flow behavior can also be expressed by the Zener–Holloman (*Z*) parameter, which is given by Equation (5):(5)$$Z=A{\left[\sinh(\alpha \sigma )\right]}^{n}=\dot{\varepsilon }\exp\left(\frac{Q}{RT}\right)$$

Taking the natural logarithm on both sides of Equation (5), Formula (6) is obtained:(6)lnZ=lnA+nln[sinh(ασ)]

Figure 7c shows the results of the linear regression of ln *Z* and ln [sinh(*ασ*)]. The slope, *n*, and intercept ln, *A*, of the experimental alloy were 2.944 and 21.696, respectively, and the corresponding *A* value was 7.09 × 10^9^. B substituting the calculated parameter values (*n*, α, *Q*, and *A*) into Equation (1), the hot deformation constitutive equation of the Mg-16Al Mg alloy can be obtained.

As shown in Figure 7c, it can be clearly found that the change of peak stress, *σ_p_*, and *Z* value exhibited a good linear relationship, and the *Z* value increased with the increase of the peak stress value, *σ_p_*, which indicated that the constitutive equation established was effective. The equation describing the relationship between the peak stress, *σ_p_*, and the *Z* parameter is as follows:(7)$$\sigma_{p}=\frac{1}{0.0189}\ln\{{\left(\frac{Z}{7.09\times 10^{9}}\right)}^{\frac{1}{2.984}}+{\left[{\left(\frac{Z}{7.09\times 10^{9}}\right)}^{\frac{2}{2.984}}+1\right]}^{\frac{1}{2}}\}$$
(8)$$Z=\dot{\varepsilon }\exp(\frac{143.989}{RT})$$

#### 3.4.1. Compensation of Strain

The influence of strain on the thermal deformation behavior of metallic materials is considered (in general) negligible, and, hence, the influence of strain is neglected in Equation (1). However, many studies have shown that the strain variable has a significant effect on the deformation activation energy and material constants in the entire strain range [31,32,33,34,35,36]. To further explore the flow deformation behavior of the Mg-16Al Mg alloy, *α*, *n*, *Q*, and ln*A* values were obtained under different strains based on experimental data. Figure 8 shows the *α*, *n*, *Q*, and ln*A* values of the experimental alloy as a function of the strain. Each value varied significantly with the strain. Therefore, the effect of strain must be considered in order to obtain a constitutive equation that accurately describes the thermal deformation behavior. To incorporate the influence of strain into the equation, the activation energy, *Q*, and the material constants (*α*, *n*, and ln*A*) are assumed to be polynomial functions of the strain [25]. 

According to the literature [20], the sixth-degree polynomial functions between the material constants and the true strain are given as follows:(9)$$\left\{ \begin{array}{l} \alpha =A_{0}+A_{1}\varepsilon +A_{2}\varepsilon^{2}+A_{3}\varepsilon^{3}+A_{4}\varepsilon^{4}+A_{5}\varepsilon^{5}+A_{6}\varepsilon^{6}\\ n=B_{0}+B_{1}\varepsilon +B_{2}\varepsilon^{2}+B_{3}\varepsilon^{3}+B_{4}\varepsilon^{4}+B_{5}\varepsilon^{5}+B_{6}\varepsilon^{6}\\ Q=C_{0}+C_{1}\varepsilon +C_{2}\varepsilon^{2}+C_{3}\varepsilon^{3}+C_{4}\varepsilon^{4}+C_{5}\varepsilon^{5}+C_{6}\varepsilon^{6}\\ \ln A=D_{0}+D_{1}\varepsilon +D_{2}\varepsilon^{2}+D_{3}\varepsilon^{3}+D_{4}\varepsilon^{4}+D_{5}\varepsilon^{5}+D_{6}\varepsilon^{6} \end{array}\right.$$

The polynomial functions (9) were fitted based on the material constants and activation energies obtained under different true strain conditions. The coefficients of the sixth-order polynomial functions are shown in Table 2, and the fitting curves are shown in Figure 8. As shown in the figure, the experimental data exhibited good correlation with the material constant obtained via polynomial function fitting. After the deformation activation energy and material constants are determined from the fitting function, the flow stress under specific strain conditions can be predicted as follows:(10)$$\sigma =\frac{1}{\alpha (\varepsilon )}\ln\left\{{\left(\frac{Z(\varepsilon )}{A(\varepsilon )}\right)}^{1/n(\varepsilon )}+{\left[{\left(\frac{Z(\varepsilon )}{A(\varepsilon )}\right)}^{2/n(\varepsilon )}+1\right]}^{1/2}\right\},\;Z=\dot{\varepsilon }\exp\left(\frac{Q}{RT}\right)$$

#### 3.4.2. Verification of Constitutive Equation

The predicted values and the measured values of the Mg-16Al experimental alloy under different conditions are compared in Figure 9, and the curves are the true stress–strain curves obtained by the experiment. As the figure shows, the predicted stress values concurred (in general) with the experimental values. However, at low temperature and high strain rate (523 K/0.1 s^−1^), the predicted values were significantly smaller than the actual values, leading to the failure of the constitutive equation. This may be attributed to two factors, namely (i) numerous twin structures are generated in the experimental alloy during deformation, as shown in Figure 18a, and (ii) the permanent microscopic strength of the Mg_17_Al_12_ phases varied significantly with the temperature, as shown in Figure 10 [37]. Under this deformation condition, many *β*-Mg_17_Al_12_ phases, which were undissolved in the matrix, became attached to regions near the twin boundaries. However, the *γ*-Mg_17_Al_12_ phases, which were dissolved in the *α*-Mg matrix, were re-precipitated from the matrix. At this time, the microscopic strength value of the Mg_17_Al_12_ phase (420 MPa) was considerably greater than the strength of the experimental alloy. These may have resulted in an increase in the overall stress values of the alloy under the condition of low temperature and high strain rate. Consequently, the predicted stress values were lower than the experimental values.

The accuracy of the constitutive equation was further evaluated, and the predictive power of this equation was quantified. This was achieved by calculating, from Equations (11) and (12) [33], the correlation coefficient (*R*) and the average absolute relative error (AARE, *E_R_*) between the experimental and predicted flow stress values:(11)$$R=\frac{\sum_{i=1}^{N}\left(E_{i}-\bar{E})(P_{i}-\bar{P}\right)}{\sqrt{\sum_{i=1}^{N}{\left(E_{i}-\bar{E}\right)}^{2}\sum_{i=1}^{N}{\left(P_{i}-\bar{P}\right)}^{2}}}$$
(12)$$E_{R}=\frac{1}{N}\sum_{i=1}^{N}\left|\frac{E_{i}-P_{i}}{E_{i}}\right|\times 100\%$$
where *E* is the experimental flow stress value, and *P* is the flow stress value predicted by the constitutive equation. Moreover, $\bar{E}$ are $\bar{P}$ the average values of experimental and predicted flow stress, respectively. *N* is the total number of the data in the study.

The correlation coefficient, *R*, represents the linear relationship strength between the experimental and the predicted stress values. The predicted value of the model may be higher or lower (than the actual value), but the *R* value increases non-monotonically with the goodness of the fit [38,39]. Therefore, the unbiased statistical parameter *AARE* was also used in this study, to verify the predictability of the model. *R* and *AARE* values of 0.9565 and 10.6%, respectively, were calculated for large-scale deformation conditions (see Figure 11a). This reflected the good correlation between the experimental data and the predicted data, and a good predictive ability of the proposed constitutive equation was noted. However, a maximum *AARE* value of 33.27% was calculated for deformation conditions such as low temperature and high strain rate (523 K/0.1 s^−1^). This value indicated that the constitutive model was inapplicable and the equation was invalid. The predicted stress values of the constitutive model were considerably smaller than the experimental stress values, and, therefore, a stress value (Δσ) was added to Equation (10), for improved stress prediction. A comparison of the predicted stress value with the experimental stress value revealed a value of 50 MPa (see Equation (13)) for the constitutive equation corresponding to this condition). Figure 11b shows the comparison between the experimental flow stress values and the predicted stress values after inclusion of the additional stress. Under all deformation conditions, the calculated *R* and *AARE* values were 0.9749 and 9.04%, respectively. After the stress value was added, the flow stress at a temperature of 523 K and a strain rate of 0.1 s^−1^ was accurately predicted by the constitutive equation (i.e., Equation (13)).
(13)$$\sigma =\frac{1}{\alpha }\ln\left\{{\left(\frac{Z}{A}\right)}^{1/n}+{\left[{\left(\frac{Z}{A}\right)}^{2/n}+1\right]}^{1/2}\right\}+50,\;Z=0.1\exp\left(\frac{Q}{4848.222}\right)$$

### 3.5. Critical Conditions of DRX

Figure 12 shows the typical strain hardening rate (*θ* = d*σ*/d*ε*) versus flow stress (*σ*) obtained at 673 K and 0.1 s^−1^. The *θ*-*σ* diagrams can be divided into three distinct stages. The first stage lasted from the initial stress to the critical stress (*σ_c_*), where the value of *θ* decreased sharply. The second stage encompassed the deformation from DRX (corresponding to *σ_c_*) to the peak stress (*σ_p_*). At this time, DRX began inside the material, and the strain hardening rate decreased with increasing flow stress, until the peak flow stress (corresponding to *σ_p_*) was reached. At *σ_p_*, the work hardening effect and dynamic recrystallization softening effect were balanced. The third segment lasted from *σ_p_* to the steady-state flow stress (corresponding to *σ_s_*), where recrystallization softening played a dominant role. At this time, the flow stress began to decrease, and the strain hardening rate decreased to a negative value. Upon rebalancing of the recrystallization softening and work hardening processes, the material entered a steady-state flow stage [40].

Poliak and Jonas [10] proposed a method of determining the critical condition required for DRX by considering the inflection point of strain hardening rate *θ*–*σ* (*θ* = d*σ*/d*ε*) relation curves. For accuracy of the calculation, the second derivative method was used to obtain the minimum values of (−d*θ*/d*σ*)–*σ* relation curves as the critical stress values of DRX, and the corresponding critical strain values were calculated. Figure 13 shows the relation curves between the *θ* and *σ* of Mg-16Al Mg alloy under certain deformation conditions. In this study, the critical strain and critical stress values under different deformation conditions were determined via the second derivative method.

The *σ_c_* and *ε_c_* of DRX are affected by both the temperature and the strain rate (as shown in Figure 14). For a given strain rate, the critical condition values decreased with increasing temperature. The diffusion rate of atoms increased with increasing deformation temperature, and the dislocations were prone to slip and climb. Moreover, the propensity for grain boundary slip increased, which was conducive to the occurrence of DRX. For a given temperature, the critical condition values increased with increasing strain rate. The time for dislocation movement was insufficient under high strain rate conditions, and, hence, elimination of the internal stress in the grains was difficult, resulting in aggregation of the dislocations. Similarly, the nucleation and growth time of recrystallized grains was also insufficient, which was unfavorable for the occurrence of DRX.

The critical stress values for DRX of the Mg-16Al Mg alloy were lower than that of other AZ-based Mg alloys [16,17], indicating that several Mg_17_Al_12_ phases hindered grain boundary migration. The consequent grain growth inhibition yielded uniform and fine grains. Moreover, many Mg_17_Al_12_ phase particles obstructed dislocation movement, causing dislocation accumulation, and the accumulation sites served as DRX nucleation points. This promoted the nucleation of recrystallized grains, and these regions were prone to DRX.

To further illustrate the effect of strain rate and temperature on the *ε_c_* of DRX, the Sellars model [14] was introduced to characterize the *ε_c_* model:(14)$$\varepsilon_{c}=aZ^{b}$$
where *a* and *b* are constants and *Z* is the Zener–Hollomon parameter. The average activation energy *Q* of Mg-16Al Mg alloy under different deformation conditions was 143.99 KJ/mol. The logarithm of both sides of Equation (14) is taken as follows:(15)$$\ln\dot{\varepsilon }=\ln a+b\ln Z$$

The *ε_c_*, *Q*, and microstructure of the experimental alloy were analyzed. For different strain rates, the single-factor linear regression of ln*ε_c_*-ln*Z* was performed (see Figure 15a), and Equation (16) described the critical strain prediction model of the Mg-16Al Mg alloy. The results of the model revealed that the high deformation temperature reduced the *Z* parameter and the critical strain, *ε_c_*, of DRX, thereby promoting the occurrence of DRX. However, the high strain rate led to an increase in the *Z*-parameter, and, hence, the *ε_c_* required the occurrence of DRX, thereby suppressing the occurrence of DRX.

Univariate linear regression was performed on the *ε_c_* versus peak strain (*ε_p_*) curves of the Mg-16Al Mg alloy (see Figure 15b for the linear relationship). The linear regression results of the *ε_c_-ε_p_* curves can be described by Equation (17), which showed that DRX occurred before the peak strain was reached.
(16)$$\varepsilon_{c}=1.32\times 10^{-4}Z^{0.18643}$$
(17)$$\varepsilon_{c}=0.6567\varepsilon_{p}-0.00631$$

### 3.6. Hot Processing Map

Processing maps based on dynamic material model theory are considered to be effective methods for optimizing the hot working parameters of many metals and alloys over a wide range of temperatures and strain rates. In this model, the workpiece is regarded as an energy dissipator, and the instantaneous power during plastic deformation is given as follows [41]:(18)$$P=\int_{0}^{\dot{\varepsilon }}\sigma d\dot{\varepsilon }+\int_{0}^{\sigma }\dot{\varepsilon }d\sigma =G+J$$
where *σ* is the flow stress, and $\dot{\varepsilon }$ is the strain rate. The first integral is expressed in *G*, which represents the energy consumed by plastic deformation of materials, most of which is converted into heat energy, and a small part is stored in the form of crystal defect energy. The second integral is expressed in *J*, which represents the energy consumed by the evolution of microstructure during the plastic deformation of materials. The distribution ratios of *G* and *J* can be described by the strain rate sensitivity index, *m*, of flow stress:(19)$$m=\frac{dJ}{dG}=\frac{\dot{\varepsilon }d\sigma }{\sigma d\dot{\varepsilon }}=\frac{d\log\sigma }{d\log\dot{\varepsilon }}$$

According to *m*, the dimensionless power dissipation efficiency, *η*, relative to microstructural changes, can be obtained as follows:(20)$$\eta =\frac{J}{J_{\max}}=\frac{2m}{m+1}$$

The power dissipation diagram is the contour map of the power dissipation efficiency, *η*, drawn on the ln$\dot{\varepsilon }$-T two-dimensional plane under a certain strain. The metallographic observation can be used to analyze the deformation mechanism of different regions with the power dissipation efficiency map. Based on the extreme principles of irreversible thermodynamics, the continuous instability criterion is defined as follows [42]:(21)$$\xi \left(\dot{\varepsilon }\right)=\frac{\partial \ln\left[m/\left(m+1\right)\right]}{\partial \ln\dot{\varepsilon }}+m\leq 0$$

The instability map consisting of $\xi \left(\dot{\varepsilon }\right)$, ln$\dot{\varepsilon }$, and T, and when the unstable parameter $\xi \left(\dot{\varepsilon }\right)$ is negative, indicating that the rheological instability such as wedge cracking, localized deformation, shear deformation, and so on has occurred. 

According to Equations (20) and (21), *η* and $\xi \left(\dot{\varepsilon }\right)$ values under different strain conditions can be obtained, and the hot processing maps can be obtained by superposing the power dissipation diagrams and instability diagrams. The hot processing maps under different strains are shown in Figure 16.

The contour lines in the hot processing map represent the value of energy dissipation efficiency, the yellow area represents the transition region (change-over region), and the red area represents the instability area (INS). The value of power dissipation efficiency greater than 30% is suitable for hot deformation of Mg alloys [43]. Generally speaking, the dissipation efficiency increases with the increase of temperature and the decrease of strain rate. Similarly, the dissipation efficiency of Mg-16Al Mg alloy increases with increasing temperature. However, when the strain rate ranges from 0.001 to 0.1 s^−1^, the values of power dissipation efficiency increase with the increase of the strain rates. While the strain rate ranges from 0.0001 to 0.001 s^−1^, the values of power dissipation efficiency decrease with the increase of the strain rates.

The hot processing maps of the Mg-16Al Mg alloy under different strain conditions are shown in Figure 16. Four different areas can be clearly distinguished, namely a workability region (Domain Ι), a metastable workability region (Domain ΙΙ), a transition region (change-over region), and an instability region (INS). The area of the change-over region and the INS increased significantly with an increasing amount of strain. Figure 17 shows the microstructure of the alloy under different deformation conditions in the workability region. These results confirmed that microstructural evolution was correlated with the value of the power dissipation efficiency. In Domain I (633–673 K and 0.001–0.1 s^−1^), the peak value of the efficiency was ~45%. The high deformation temperature enhanced thermal activation and promoted grain boundary sliding. Furthermore, the initiation of non-basal slip systems and the increasing dislocation density during the thermal deformation process facilitated the nucleation of DRX. In addition, the movement of dislocations was hindered by many *β*-Mg_17_Al_12_ phases distributed in a discontinuous network at the GBs and the Mg_17_Al_12_ phases precipitated from the *α*-Mg matrix. This resulted in the accumulation of dislocations, and the accumulation sites became the nucleation sites for DRX. Figure 17 shows the microstructure after deformation at 673 K and different strain rates. The microstructure consisted of a uniform equiaxed structure, with a small amount of elongated *β*-Mg_17_Al_12_ phase sandwiched at the GBs. With decreasing strain rate, the DRX grains had sufficient time to grow, and the average grain size increased from 12 μm at a strain rate of 0.1 s^−1^ to 22 μm at a strain rate of 0.001 s^−1^. Due to the large number of Al atoms in the experimental alloy, Mg_17_Al_12_ phases were inevitably precipitated at the GBs and inside the grains. Decreasing strain rate led, however, to a reduction in the precipitation of these phases from the matrix within the GBs and grains (except for the discontinuous bulk *β*-Mg_17_Al_12_ phases). At a strain rate of 0.001 s^−1^, only a small amount of Mg_17_Al_12_ phases precipitated at the GBs.

Domain II was characterized by a peak dissipation efficiency value of 60% (523–623 K and 0.01–0.1 s^−1^). Unfortunately, although the dissipation value in this region was high and increased with increasing strain rate, the deformed specimen will generate high local viscosity heat, due to the high strain rate. This may lead to the generation of local shear bands and microcracks. Figure 18a shows the plastic deformation at a temperature of 523 K and a strain rate of 0.1 s^−1^. Considerable dislocation proliferation and entanglement occurred during the deformation process. The corresponding formation of numerous *β*-Mg_17_Al_12_ phases increased the hindrance to dislocation motion, resulting in a large stress concentration at the GBs and, consequently, crack initiation. In addition, at low temperatures (<573 K), basal slip was the dominant deformation mechanism, with only a small amount of cylindrical slip and conical slip occurring along favorable orientations. During deformation, twin nucleation was promoted, and the number of sub-crystals increased. The coordinated plastic deformation of these crystals in subsequent deformation may lead to a chain-like recrystallization structure (see Figure 18a), and partially DRX grains appeared near the *β*-Mg_17_Al_12_ phases and the GBs. With increasing temperature, dislocation climb and slip became easier than at lower temperatures. The increase in dislocation density during deformation facilitated the nucleation of DRX. Many fine DRX grains appeared in the microstructure, as shown in Figure 18b. When the deformation temperature reached 623 K, the *σ_c_* of non-basal surface slip decreased, the degree of stress concentration was significantly reduced, and the atomic thermal activation capacity was enhanced. At this time, each sliding system, especially the non-basal surface sliding system, became the dominant means of releasing stress concentration and coordinating plastic deformation. The Mg-16Al Mg alloy underwent continuous DRX at a temperature of 623 K and a strain rate of 0.1 s^−1^, as shown in Figure 18c. The resulting microstructure consisted of fine equiaxed crystals and several discontinuous Mg_17_Al_12_ phases that were distributed at the GBs.

The yellow part in the processing map corresponded to the change-over region (523–637 K and 0.001–0.01 s^−1^; peak power dissipation efficiency: ~30%). The optical micrographs and the SEM images are shown in Figure 19 and Figure 20, respectively. Microstructures undergoing DRX were observed at 573 K and strain rates of 0.01 and 0.001 s^−1^, respectively. A small amount of massive discontinuous *β*-Mg_17_Al_12_ phases was present in these microstructures. Moreover, many discontinuous fine Mg_17_Al_12_ phases were reprecipitated in the recrystallized grain boundaries and even within the grains. With decreasing strain rate, the average recrystallization grain size increased from 2 to 4 μm. Deformation conditions of 623 K and 0.001 s^−1^ yielded a fully recrystallized coarse-grained microstructure consisting mainly of equiaxed crystals (Figure 19c). The recrystallized grain size increased from 2 μm at 0.1 s^−1^ to 12 μm, and the discontinuous massive *β*-Mg_17_Al_12_ phases were distributed on triangular grain boundaries. As shown in Figure 20c, a small amount of dot-like Mg_17_Al_12_ phases precipitated inside the recrystallized grains.

A distinct instability region (523–633K and 0.0001–0.001 s^−1^) was observed in the processing map. As shown in Figure 21, when hot deformation occurred at a low strain rate, each deformed specimen underwent complete DRX (irrespective of the temperature), and the size of the recrystallized grains increased with increasing temperature. The dissipation efficiency was high (peak value: 50%) in the INS, owing to the occurrence of complete DRX. Due to the low strain rate, sufficient time for Al-atoms diffusion, and the high Al content of the alloy specimens, several *γ*-Mg_17_Al_12_ phases dissolved in the experimental alloy were re-precipitated from the α-Mg matrix.

Figure 22a shows SEM images of the specimens deformed at 523 K and 0.0001 s^−1^. As the results showed, the Mg_17_Al_12_ phases precipitated from the recrystallized grains, and the discontinuous *β*-Mg_17_Al_12_ phases, which were undissolved in the matrix, occurred throughout the entire specimen microstructure. This may have resulted in the high dissipation efficiency. For deformation at temperatures above 573 K, the sizes of the recrystallized grains increased with increasing temperature. Furthermore, numerous *γ*-Mg_17_Al_12_ phases precipitated from the recrystallized grains, as shown in Figure 21b,c. Figure 21d shows that, for deformation conditions of 673 K and 0.0001 s^−1^, the recrystallized grains grew significantly and were elongated along the direction perpendicular to the compression direction. Moreover, except for the bulk *β*-Mg_17_Al_12_ phases that were undissolved in the matrix, the *γ*-Mg_17_Al_12_ phases were discontinuously precipitated inside the grains. The INS was observed by means of SEM, as shown in Figure 22. Many microcracks occurred in this region. The generation of numerous Mg_17_Al_12_ phases, microcracks, and vacancies may lead to an increase in the dissipation efficiency. Therefore, hot working in this unstable region should be avoided.

The hot processing maps and microstructure of the Mg-16Al Mg alloy were analyzed, and this analysis revealed that a high Al content was unfavorable for the plastic forming during hot working. Moreover, compared with other areas, Domain I, which consisted of uniformly sized DRX grains, was associated with high dissipation efficiency and was more suitable for hot working. The window for hot deformation applying the range of test parameters considered was very narrow, as indicated by temperature and strain rate ranges of 633–673 K and 0.001–0.1 s^−1^, respectively.

## 4. Conclusions

In the present study, the hot compression deformation behaviors and processing maps of an extruded Mg-16Al Mg alloy bar after solution treatment were investigated under various strain rate (0.0001–0.1 s^−1^) and temperature (523–673 K) conditions. The major findings are summarized as follows:The original extruded microstructure of the Mg-16Al alloy was mainly composed of α-Mg grains, *β*-Mg_17_Al_12_ phases with discontinuous reticular distribution at the GBs, and γ-Mg_17_Al_12_ phases with a cellular structure inside the α-Mg grain boundaries. After the solution treatment, the discontinuous *β*-Mg_17_Al_12_ phases were partially dissolved, whereas the *γ*-Mg_17_Al_12_ phases were completely dissolved in the α-Mg matrix. Massive *β*-Mg_17_Al_12_ phases persisted at the GBs.The Mg-16Al solid solution alloy underwent significant work hardening and continuous flow softening at relatively high strain rate levels ($\dot{\varepsilon }$≥ 0.01 s^−1^). However, at strain rates lower than these levels, the initial work hardening components of the alloy decreased at high temperatures (T ≥ 623 K) and the flow softening appeared as steady-state flow. Nevertheless, at temperatures lower than 623 K, the initial work hardening occurred only at relatively low strains.The precipitation of numerous Al-containing second phase particles during the deformation process of the alloy hindered the movement of dislocations and promoted the nucleation of DRX. Consequently, the critical strain, ε_c_, of DRX was less than that of the common commercial-grade Mg alloys comprising the AZ series.The relatively large number of Al atoms in the Mg-16Al solid solution alloy promoted the nucleation of DRX and refinement of the grains. Similarly, the coordinated interaction between the diffusion and migration of these atoms and the dislocation movement during hot working deformation reduced the steady-state deformation range. For deformation at high temperatures and low strain rates, the slow dissolution of Mg_17_Al_12_ phases stimulated the initiation of local microcracks.The suitable hot working range of the Mg-16Al Mg alloy after solution treatment was narrow: T = 633–673 K, $\dot{\varepsilon }$ = 0.001–0.1 s^−1^.

## Figures and Tables

**Figure 1 materials-13-03107-f001:**
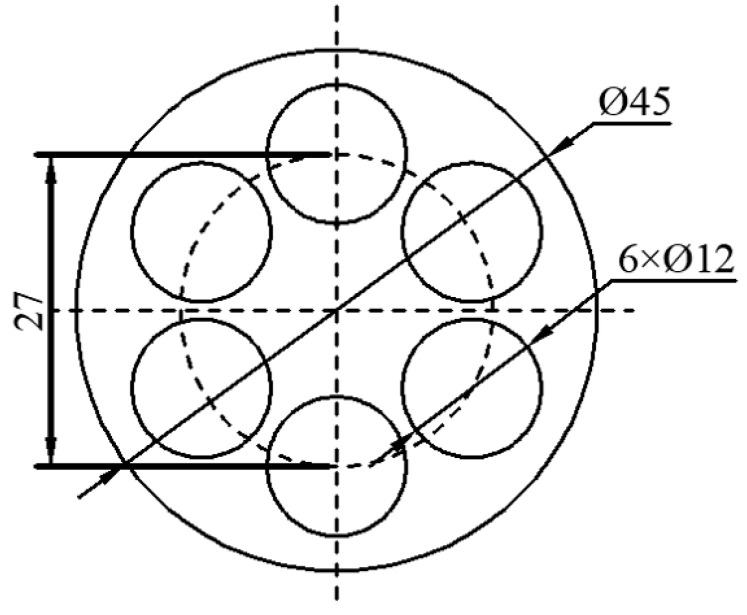
Sampling location of hot compressed specimens.

**Figure 2 materials-13-03107-f002:**
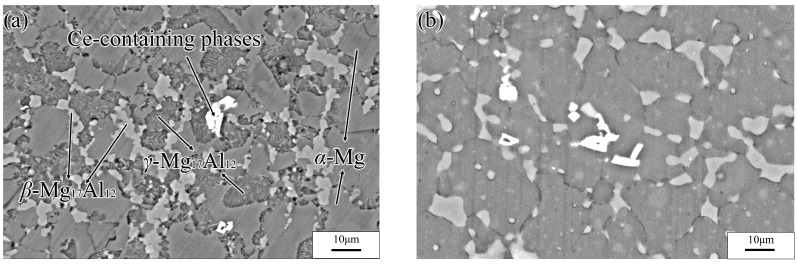
SEM images of Mg-16Al Mg alloy before (**a**) and after (**b**) solution treatment with 400 °C holding for 12 h.

**Figure 3 materials-13-03107-f003:**
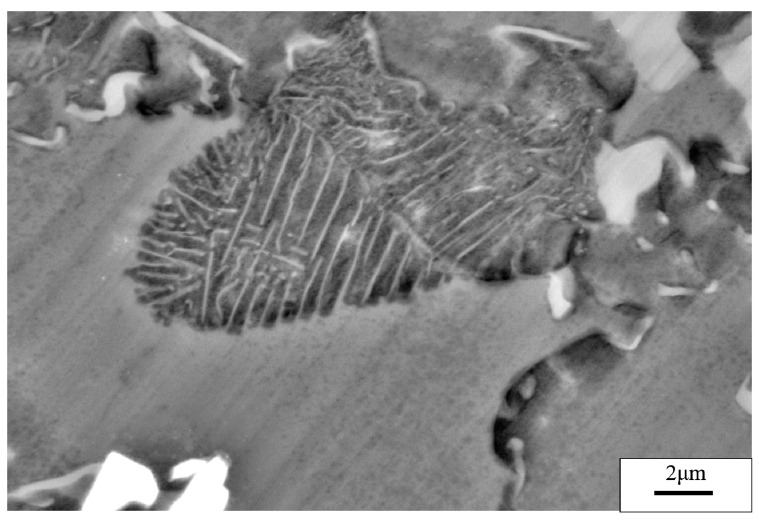
SEM image of the *γ*-Mg_17_Al_12_ phase.

**Figure 4 materials-13-03107-f004:**
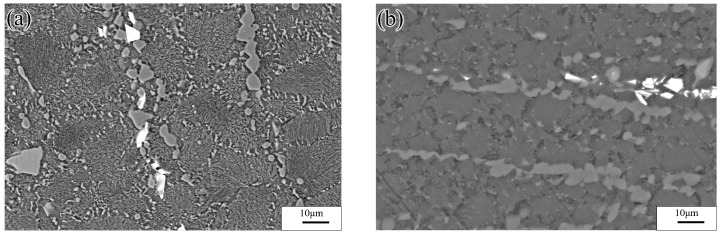
SEM images of experimental alloy deformed under different conditions: (**a**) 523 K/0.0001 s^−1^ and (**b**) 673 K/0.1 s^−1^.

**Figure 5 materials-13-03107-f005:**
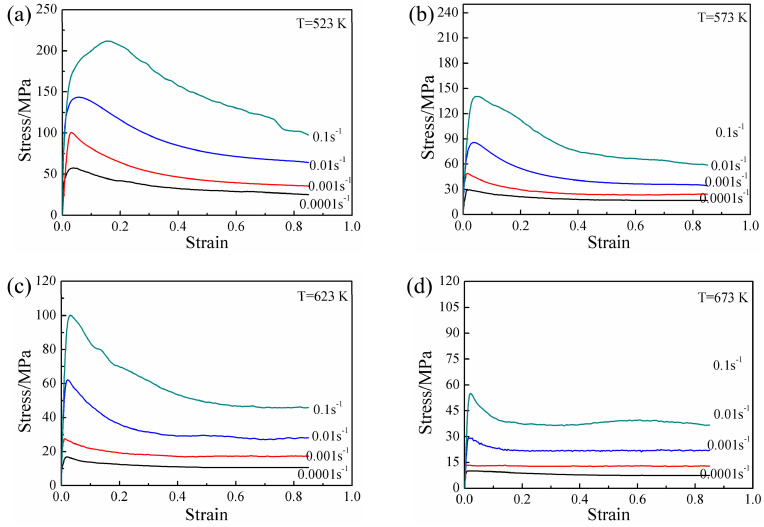
True stress–strain behavior of experimental alloy obtained by hot compression tests at different temperatures: (**a**) 523 K, (**b**) 573 K, (**c**) 623 K, and (**d**) 673 K.

**Figure 6 materials-13-03107-f006:**
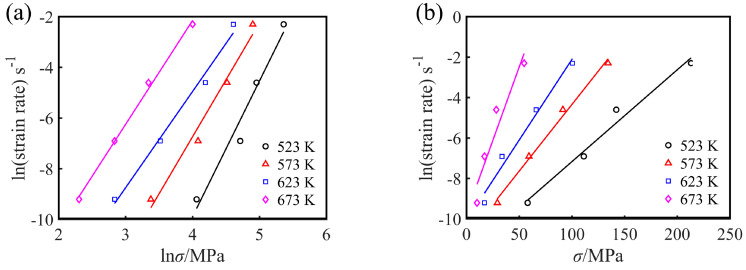
(**a**) ln$\dot{\varepsilon }$-ln*σ* and (**b**) ln$\dot{\varepsilon }$-*σ* linear regression curves for the experimental alloy.

**Figure 7 materials-13-03107-f007:**
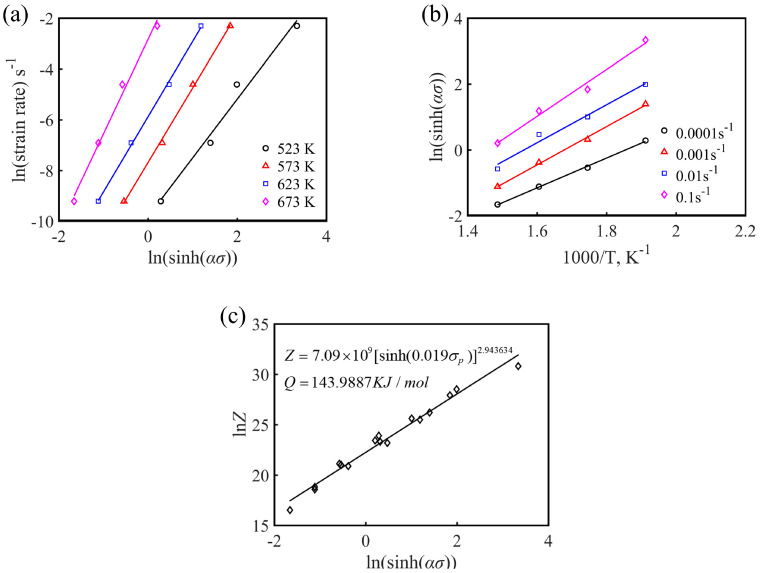
(**a**) ln$\dot{\varepsilon }$-ln[sinh(*ασ*)], (**b**) ln[sinh(*ασ*)]-1000/T, and (**c**) ln*Z*-ln[sinh(*ασ*)] linear regression relationship for the experimental alloy.

**Figure 8 materials-13-03107-f008:**
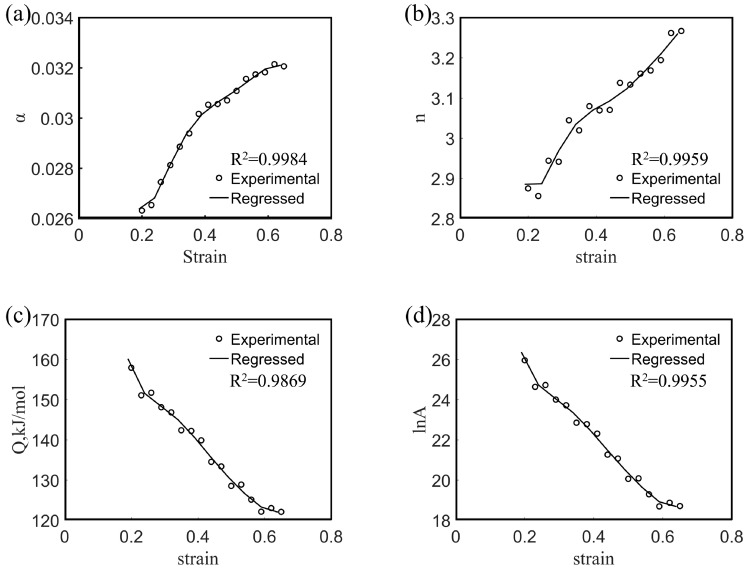
Variations of *α* (**a**), *n* (**b**), *Q* (**c**), and ln*A* (**d**) with true strain based on sixth-order polynomial-fit for the experimental alloy.

**Figure 9 materials-13-03107-f009:**
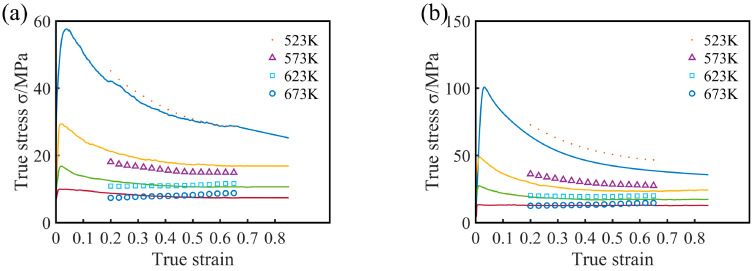
Comparisons between predicted and measured flow stress curves of Mg16Al alloy at different strain rate: (**a**) 0.0001 s^−1^, (**b**) 0.001 s^−1^, (**c**) 0.01 s^−1^, and (**d**) 0.1 s^−1^.

**Figure 10 materials-13-03107-f010:**
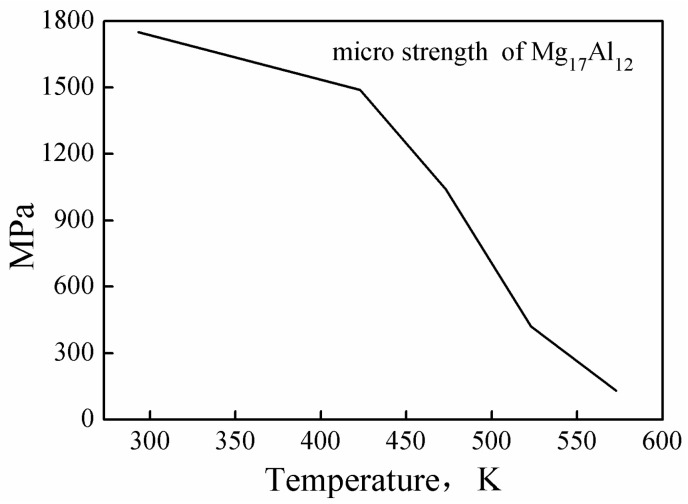
Permanent microscopic strength values of the Mg_17_Al_12_ phase at different temperatures. [37].

**Figure 11 materials-13-03107-f011:**
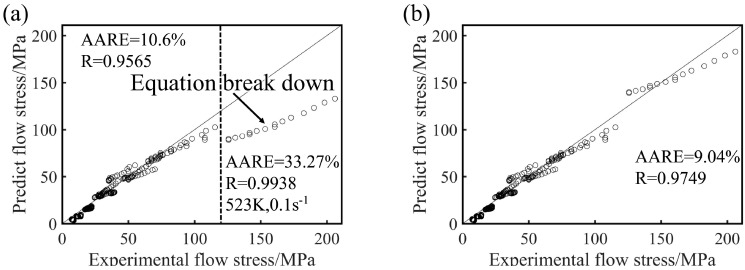
The comparison between the experimental flow stress and the flow stress predicted by the constitutive equation before (**a**) and after (**b**) inclusion of the additional stress.

**Figure 12 materials-13-03107-f012:**
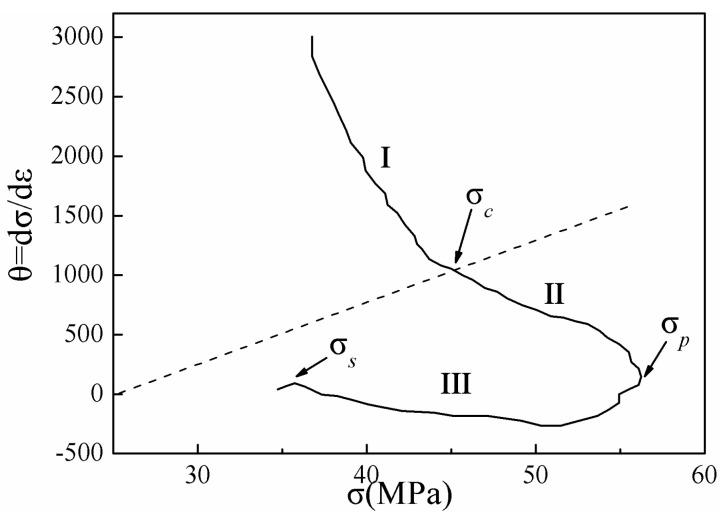
Typical *θ*–*σ* plot obtained at 673 K and 0.1 s^−1^.

**Figure 13 materials-13-03107-f013:**
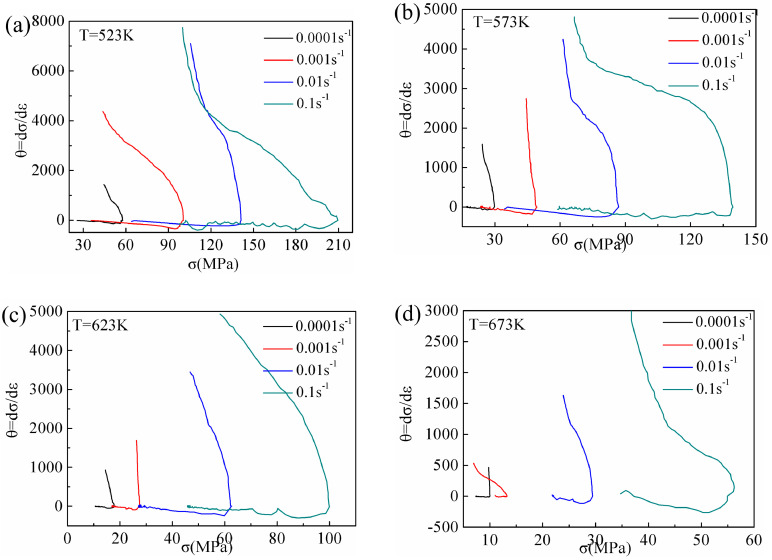
*θ*-*σ* curves at different deformation conditions: (**a**) T = 523 K, (**b**) T = 573 K, (**c**) T = 623 K, and (**d**) T = 673 K.

**Figure 14 materials-13-03107-f014:**
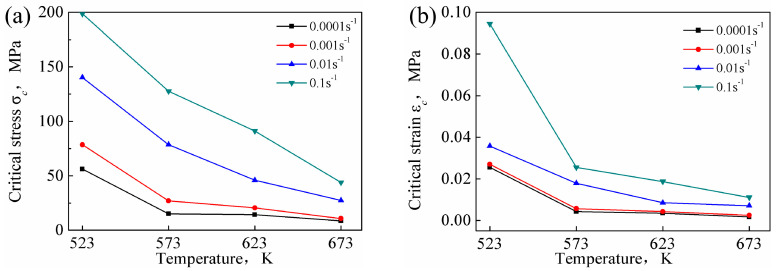
*σ*_c_-T curves (**a**) and *ε**_c_*-T curves (**b**) at different deformation conditions.

**Figure 15 materials-13-03107-f015:**
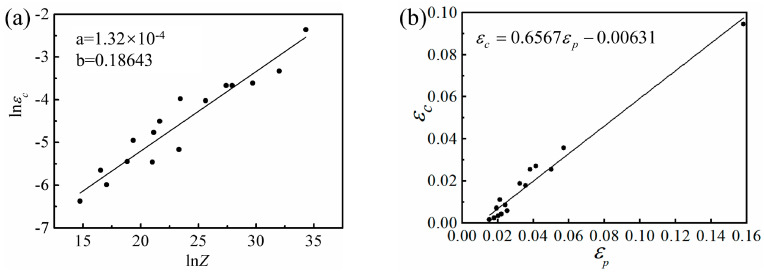
ln*ε_c_*-ln*Z* relationship curve (**a**) and *ε_p_*–*ε_c_* relationship curve (**b**).

**Figure 16 materials-13-03107-f016:**
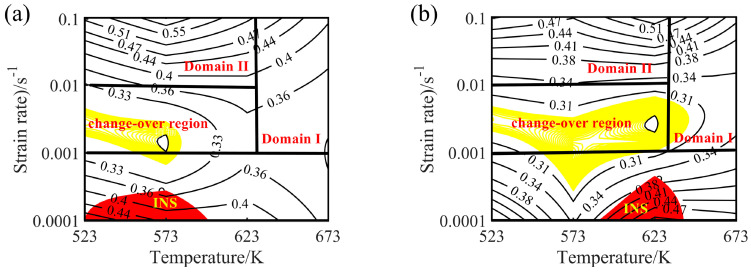
Processing maps for the experimental alloy at different true strains: (**a**) 0.2, (**b**) 0.3, (**c**) 0.4, and (**d**) 0.6. Four different areas can be distinguished, namely a workability region (Domain Ι), a metastable workability region (Domain ΙΙ), a transition region (change-over region), and an unsuitable workability region, i.e., instability region (INS).

**Figure 17 materials-13-03107-f017:**
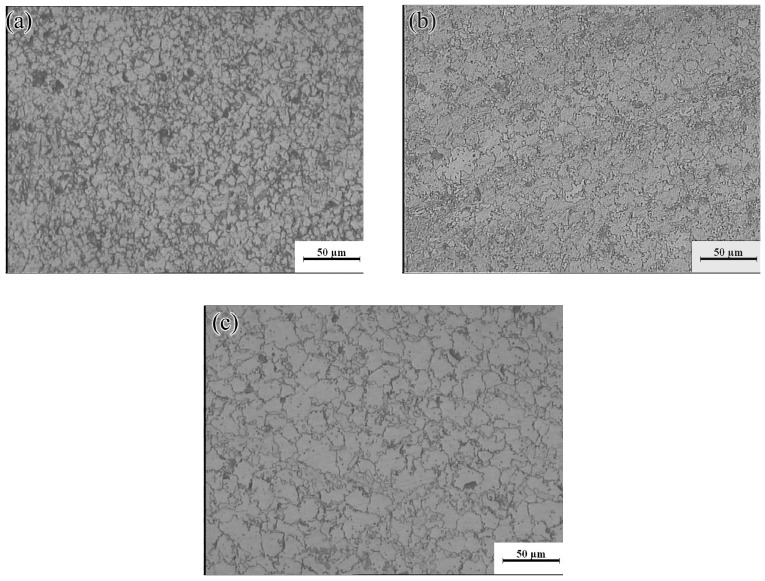
Microstructures of the Mg-16Al Mg alloy in Domain I: (**a**) 673 K/0 0.1 s^−1^, (**b**) 673 K/0 0.01 s^−1^, and (**c**) 673 K/0 0.001 s^−1^.

**Figure 18 materials-13-03107-f018:**
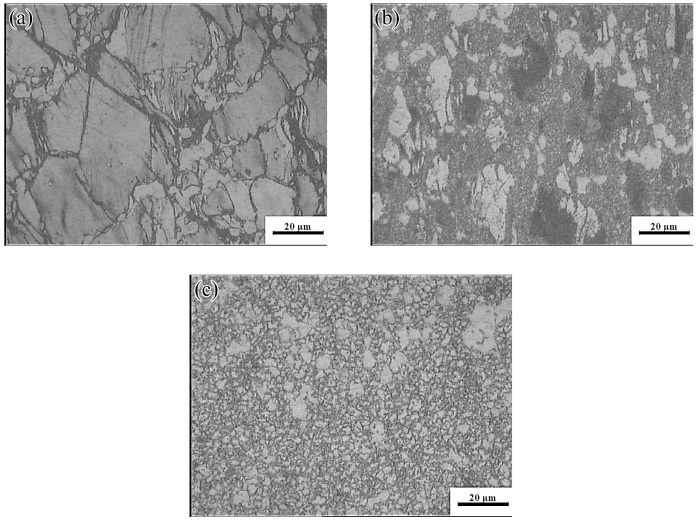
Microstructures of the Mg-16Al Mg alloy in Domain II: (**a**) 523 K/0 0.1 s^−1^, (**b**) 573 K/0 0.1 s^−1^, and (**c**) 623 K/0 0.1 s^−1^.

**Figure 19 materials-13-03107-f019:**
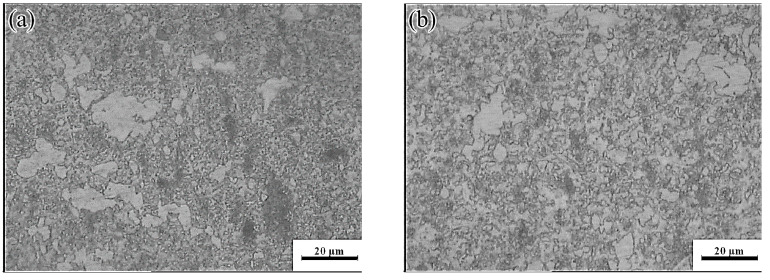
Microstructures of the Mg-16Al Mg alloy in change-over region: (**a**) 573 K/0 0.01 s^−1^, (**b**) 573 K/0 0.001 s^−1^, and (**c**) 623 K/0 0.001 s^−1^.

**Figure 20 materials-13-03107-f020:**
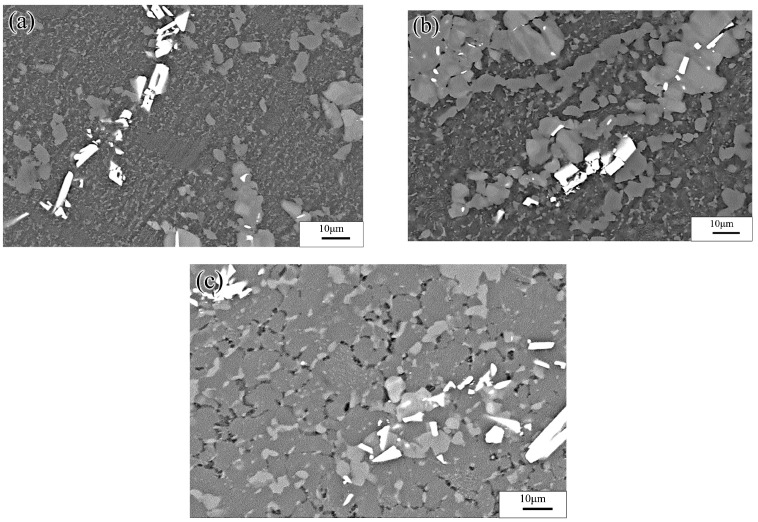
SEM images of the Mg-16Al Mg alloy in change-over region: (**a**) 573 K/0 0.01 s^−1^, (**b**) 573 K/0 0.001 s^−1^ and (**c**) 623 K/0 0.001 s^−1^.

**Figure 21 materials-13-03107-f021:**
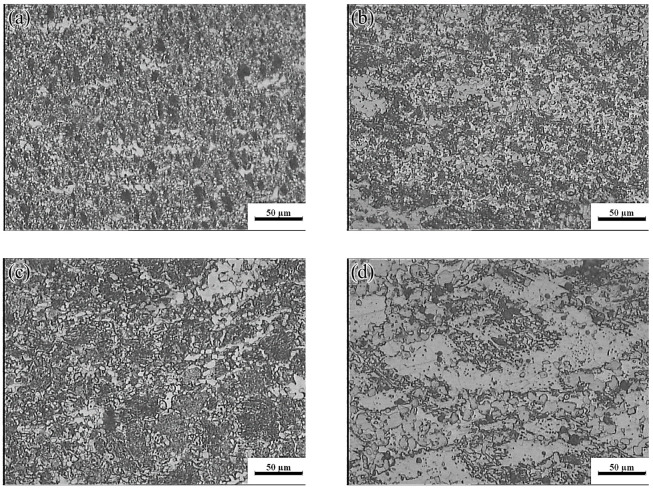
Microstructures of Mg-16Al Mg alloy in INS: (**a**) 523 K/0.0001 s^−1^, (**b**) 573 K/0.0001 s^−1^, (**c**) 623 K/0.0001 s^−1^, and (**d**) 673 K/0.0001 s^−1^.

**Figure 22 materials-13-03107-f022:**
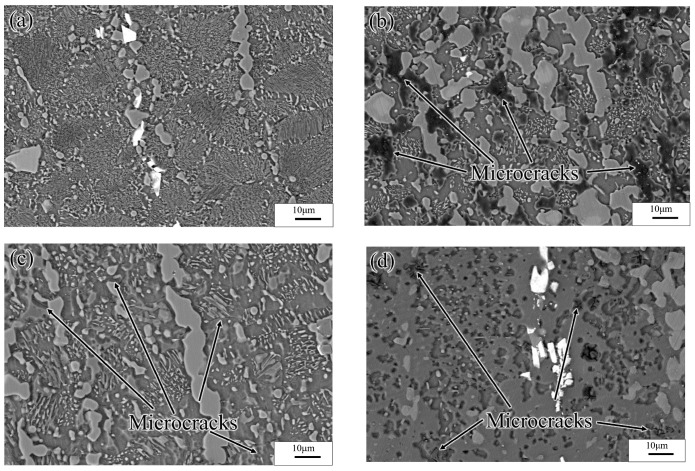
SEM images of Mg-16Al Mg alloy in INS: (**a**) 523 K/0.0001 s^−1^, (**b**) 573 K/0.0001 s^−1^, (**c**) 623 K/0.0001 s^−1^, and (**d**) 673 K/0.0001 s^−1^.

**Table 1 materials-13-03107-t001:** The *Q*, *α*, and *n* values of various extruded AZ Mg alloys.

	*Q* (KJ/mol)	*α* (MPa^−1^)	*n*
AZ41	130	0.010	4.1
AZ61	115	0.004	5.3
AZ80	105	0.004	3.3
Mg-16Al	144	0.019	3.0

**Table 2 materials-13-03107-t002:** Polynomial fitting results of *α*, *n*, *Q*, and ln*A* for the Mg-16Al alloy.

*α*	*n*	*Q*	ln*A*
A_0_ = 0.0500	B_0_ = 2.8496	C_0_ = 317.7512	D_0_ = 56.8524
A_1_ = −0.3767	B_1_ = −2.0641	C_1_ = −2266.6117	D_1_ = −439.7272
A_2_ = 2.1786	B_2_ = 5.9022	C_2_ = 12,292.0656	D_2_ = 2400.9266
A_3_ = −5.8648	B_3_ = 31.2201	C_3_ = −34,827.3340	D_3_ = −6850.4428
A_4_ = 8.2490	B_4_ = −133.3649	C_4_ = 52,452.9061	D_4_ = 10,374.5083
A_5_ = −5.8850	B_5_ = 173.2873	C_5_ = −39,917.3170	D_5_ = −7927.3370
A_6_ = 1.6898	B_6_ = −75.0813	C_6_ = 120,70.3772	D_6_ = 2403.0213

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
