# Peer review of "Constitutive Equation and Hot Processing Map of Mg-16Al Magnesium Alloy Bars"

_materials, 2020, doi:10.3390/ma13143107_

Round 1
Reviewer 1 Report
Please check all of your spellings and grammar prior to finalizing the manuscript.
A very nice contribution to the subject of processing of magnesium alloys.
Case studies of practical examples would be nice to include in the paper.
Reviewer 2 Report
Materials an Methods:
L69: delete obsolete text (from the template)
The chemical composition given in L73-74, is this as specified by the manufacturer or did you perform a chemical analysis?
Usually, we use the term »etched« where you use »corroded«. (The specimens were etched, the solution is an etchant… - please change the terminology).
Results and Discussion:
Microstructure:
SEM images: the datazone in the images is redundant adn does not contribute to the manuscript. The image should show a scalebar and that should be enough.
L107-108: it is not really clear what you mean by »precipitated along a fixed crystallographic orientation«. Please explain (or remove, if irrelevant, since you do not provide any evidence of crystallographic orientation in this manuscript).
Constituitive analysis:
L164:deformation, not deformed temperature.
The formatting of equations is strange – check the typesetting of the equations.
The diagrams are very informative and well thought. In Figure 8, is the R-value R2 or R?
Figure 9: correct y-axis title to true (and not ture) stress.
Hot processing maps:
Figure 17: the insets in the figures cover the details of the larger figures. I suggest the authors redraw the figure to not obscure any part of the images.
Reviewer 3 Report
At the end of the 15th line + beginning of the 16th line: There is “stress compensation Arrhenius equation” There should be probably “strain compensation” instead of stress compensation. Replace the text in the lines 15-16 “Combined with the stress compensation Arrhenius equation…” for “Combined with the strain compensation Arrhenius equation…”.
At the end of the 50th line + beginning of the 51st line: There is “deformation resistance” Maybe, the term of “flow stress” can be more accurate. Replace the text in the lines 50-51 “…reduce the deformation resistance, thereby…” for “…reduce the flow stress, thereby…”.
Line 69: The first part of the sentence “Materials and Methods should be described with sufficient details to allow others to replicate” don’t fit into the rest of paragraph. Probably it’s a part of instructions and should be removed. Remove the redundant text in line 69 “Materials and Methods should be described with sufficient details to allow others to replicate”.
Line 101: The term “Subsection” is redundant and should be removed. Remove the redundant word in line 101 “Subsection”.
Line 168 (equation 1) and beginning of the line 169: The term “ε̇” and “Δ should be written by the same way (i.e. dot above epsilon). In the manuscript, the symbol for the “strain rate” is sometimes written as ε̇ (e.g. equation 1) and sometimes as Î (e.g. line 169). The text of the manuscript should be checked and the symbol for the strain rate should be written uniformly, i.e. either with a dot above the epsilon or with a comma above the epsilon.
Line 274: there is a link to a figure 18a. (ii). Where is this figure? The figure 18a is placed quite far away from the line 274 (I had a little problem to locate this figure for the first time). However, this is not a big mistake and it can probably remain unchanged.
Reviewer 4 Report
The authors report a nice piece of experimental and theoretical work to define the hot workability of the Mg16Al alloy. It is usefully summarized in clear diagrams and maps.
Their account is nicely organized and clear, but there are a number of relatively minor issues which would benefit from the attention of the authors prior to any acceptance for publication.
Line 92 the word ‘burnished’ is not correct. To burnish means to polish by rubbing to deform the metal surface, forcing it into flatness. The preparation of metalloscopic samples different; it is a cutting process, in which the surface is cut, without substantial deformation by successively finer SiC abrasive powders, and finally polished by even finer cutting powders such as diamond. Additionally, the word ‘corroded’ is correct but not usual in this context. It would be usual to say ‘etched’. The authors clearly describe the composition of their etchant and their etching technique.
Fig 9 ‘Ture’ should read ‘True’.
Fig 16 is a central result for the paper. The authors are therefore invited to consider emphasising its meaning to the average reader by giving a more extended, helpful caption. For instance, the Transition Regions denoting a change from what to what; what are Domains I and II; what is INS; what do the contours represent?
The observations listed below are merely for the benefit of the authors; these comments are not necessarily recommended revisions.
It is a pity that the authors do not appear to have controlled the orientation of their microstructures, for instance, by keeping the deformation direction vertical. The evidence of directionality which can be seen in some micrographs seems random.
In Fig 8 the 6th order polynomial has to be questioned, when many experimentalists would have been delighted to achieve data which would so clearly fit to a straight line with what seems to be more than adequate accuracy.
Fig 10 This result for the strength of Mg17Al12 is, for this reviewer, difficult to believe such a fall over the range of temperature. A sudden fall at its melting point would have been believable. Is there any evidence that other factors are at work which would make the result believable?
The authors draw attention to the appearance of cracks leading to flow instability. They seem unaware of the work in the West by a prof in the UK, Campbell, who has researched extensively, finding that the cracks are caused by the melting and casting conditions, especially for the light alloys Mg and Al. They do not form as the result of stress; they merely open and become visible during deformation. This reviewer does not claim to be an expert in this field. It seems the cracks are caused by the folding over of the oxide film on the liquid, to form crack-like doubled films defects he calls bifilms. The line of Ce-rich particles which appear to be cracked in Fig 20 is typical of particles which nucleate and grow on bifilms, and which therefore appear to be cracked. The bifilm cracks can be avoided by filtering during casting. The Mg-Al alloy would not then appear to crack during deformation. The authors are recommended to consult these new findings.
